# From Reparative Surgery to Regenerative Surgery: State of the Art of Porous Hydroxyapatite in Cranioplasty

**DOI:** 10.3390/ijms23105434

**Published:** 2022-05-13

**Authors:** Ismail Zaed, Andrea Cardia, Roberto Stefini

**Affiliations:** 1Division of Neurosurgery, ASST Ovest Milanese, Legnano Hospital, 20025 Milan, Italy; roberto.stefini@gmail.com; 2Department of Neurosurgery, Neurocenter of South Switzerland, EOC—Ente Ospedaliero Cantonale, 6900 Lugano, Switzerland; andreacardia@yahoo.it

**Keywords:** hydroxyapatite, porous hydroxyapatite, cranioplasty, cranial reconstruction, decompressive craniectomy

## Abstract

Decompressive craniectomy is one of the most common neurosurgical procedures, usually performed after neuropathological disorders, such as traumatic brain injury (TBI), but also vascular accidents (strokes), erosive tumours, infections and other congenital abnormalities. This procedure is usually followed by the reconstruction of the cranial vault, which is also known as cranioplasty (CP). The gold-standard material for the reconstruction process is the autologous bone of the patient. However, this is not always a feasible option for all patients. Several heterologous materials have been created in the last decades to overcome such limitation. One of the most prominent materials that started to be used in CP is porous hydroxyapatite. PHA is a bioceramic material from the calcium phosphate family. It is already widely used in other medical specialties and only recently in neurosurgery. In this narrative review of the literature, we summarize the evidence on the use of PHA for cranial reconstruction, highlighting the clinical properties and limitations. We also explain how this material contributed to changing the concept of cranial reconstruction from reparative to regenerative surgery.

## 1. Introduction

Decompressive craniectomy (DC) is one of the most common neurosurgical procedures performed in a neurotraumatological setting, usually performed after disorders that cause significant increases in intracranial pressures, such as traumatic brain injury (TBI), but also vascular accidents (strokes), infections and other congenital abnormalities. DC, but also non-decompressive craniectomy, are usually followed by the reconstruction of the cranial vault, which is also known as a cranioplasty (CP) [1,2]. The need for CP is not only due to cosmetic reasons, but also because it has been shown that the reconstruction helps in the recovery not only of physiological blood and cerebrospinal fluid flow, but also in mental health status [3,4,5].

Currently, it is widely accepted that the first option treatment should be the autologous bone of the patient itself, since it maintains the biomimetic properties of the cranial vault and it is a cheaper option compared to other heterologous materials. However, this is not always a feasible option, since the graft could be lost during the decompressive surgery, in the case of traumatic patients, or during the storage phase [6]. Moreover, in the case of successful implantation, the autologous bone reported a high rate of postoperative complications, and, most importantly, symptomatic bone reabsorption that needed a revision surgery [3].

In order to overcome those limitations, in recent decades, several heterologous materials have been developed. The main ones used in cranioplasty are titanium, polyether-ether-ketone (PEEK), polymethyl-methacrylate (PMMA) and porous hydroxyapatite (PHA). Most of these materials have been used for several decades in the field of neurosurgery. However, they still present an important rate of postoperative complications and most importantly they do not allow effective integration of the device with the cranial vault since they are not “recognized” by the osteoblasts. More recently, PHA is the latest to be developed and caught the attention of the neurosurgical community for its biosimilarity with the autologous bone [3,4,6,7].

Here, the authors aim to present a comprehensive review of the literature concerning the development and the use of porous hydroxyapatite in cranial reconstruction, highlighting the potentialities and limitations of such material. 

## 2. Molecular Characteristics and Physicochemical Properties

Modern medical sciences, among which neurosurgery, has recognized the importance of the concept of bio-mimetic materials and it is trying to apply it to all possible aspects of everyday practice, among which the creation of the new cranioplasty implants [8]. An effective example of such scientific efforts could be seen in the creation of porous hydroxyapatite cranial implants. Those are biomimetic devices that aim to resemble the human skull.

Hydroxyapatite, a basic calcium phosphate, is one of the most present minerals in the human body, taking up to 50% by volume and 70% by weight of human bone, and today can be developed on a laboratory scale with characteristics similar to its natural counterpart. Because of its crystalline composition, it is accepted as a bioactive material. As a bioactive, it is meant that such material is able to support bone ingrowth and osteointegration, which will be further analyzed in the next section. This has been already widely proven in other medical specialties, such as orthopedic surgery [9,10,11,12]. Over the years, hydroxyapatite has been produced and used in different forms, such as powders and pastes that can be quickly used in an intraoperative setting, but also more complex porous “blocks”, in order to cover cranial defects which are present from the previous craniectomy [10,11,12,13].

In order to have an effective biomimetic action after the implantation of the device, the sole biochemical composition is not sufficient. In fact, there are other properties that neurosurgeons and manufacturers should take into consideration in the creation and application of such devices, such as the density of the material itself, the shape of the pore and the size of the pore to promote such osteointegration [14].

For what concerns the porosity, it has been shown in vivo studies that the porosity rate is fundamental in the process of osteointegration with the cranial vault since those elements influence the migration of the osteoblasts from the cranial vault to the implanted devices but also the proliferation of these cells that are involved in the process itself, such as the of osteoblasts and mesenchymal cells. Of similar importance is the shape of the pores, since a good integration is facilitated by a good ratio of matrix deposition and empty spaces [15,16]. The optimal size of the pores has been widely studied; there are studies that support an ideal dimension of 100–200 μm, since a lower dimension tends to allow the formation of fibroid tissue rather than a mineralization [15,16]. Studies have also shown the importance of the development of an interconnection pathway to achieve good integration. A structured vascularisation within the cranial implantation is the base of biological integration between the two structures. Further studies on this topic suggested that an incomplete or a minimal pore interconnection affect negatively the biomimetic and integrative process since it affects negatively the ability of the cranial blood vessels to invade the devices [17,18]. A strong vascularisation, not only in the skull but in all biological structures, is essential for the circulation not only of oxygen but also nutrients, allowing the tissue regrowth and the healing process [17]. From the physical point of view, it should be also highlighted that this grade of porosity reduces the mechanical strength of the device, increasing the risk of post-implantation fracture. As discussed below in the section about the custom-made cranioplasty, this risk is significantly reduced after the osteointegration with the cranial vault. 

From the clinical point of view, it has been recently also shown that PHA can prevent the formation of bacterial biofilms, resulting in greater resistance to infections [19]. A more recent study published by Amendola et al. that aimed to study the effectiveness of preoperative strategies to reduce the risk of postoperative complications. In their study, they suggested, based on the current evidence, that the hydrophilic nature of HA together with its porous structure and its rough surface are key factors in preventing early bacterial adhesion a proliferation, thus reducing risk of infection [20,21,22].

## 3. Radiological Evaluation and Osteointegration

The ability of porous hydroxyapatite to allow osteointegration of the implant with the bone of the patients is already known and it has been firstly shown by Martini et al. in a preclinical animal model [23]. An in vivo study conducted by Mastrogiacomo and his colleagues aiming to define the role of scaffold internal structure in bone formation showed significant histomorphometric evidence of an osteogenic process within the prosthesis itself [18]. Concerning the use of PHA cranial implant in particular, there is a large body of literature reporting an effective integration, ranging from a slightly below half (49.8%) of the clinical cohort considered in the study in the first six months from the surgical implantation, up to 74.5% in the following controls [24,25,26,27]. Staffa et al. published a series of sixty patients implanted with PHA cranioplasty where they describe their surgical technique and their results. In their follow-up analysis they reported a good integration in most of the patients; only in seven cases (11.7%) the rate of integration was suboptimal, meaning an osteointegration lower than 75% [28]. Hardy et al. tried to propose in their retrospective clinical series of eight patients a new classification of the level of integration to better define the outcome; moreover, in their study such level of integration has been reported [29]. The difficulty in analyzing and defining the osteogration has been also seen in the study published by Maenhoudt et al., where they reported the presence of a dural ossification that was difficult to analyze [27]. Further studies on the osteointegration in the pediatric population was performed Zaccaria et al., where they analyzed their cohort of six patients, finding a complete integration in five of them, with only one case of incomplete (integration of 69%) osteointegration, despite the long term follow up that ranged from 9 up to 40 months [30].

As discussed above, despite several efforts from different neurosurgical groups, there is still a discussion on how to classify and analyze the osteointegration. Maenhoudt et al. proposed an innovative system based on the Materialise Interactive Medical Image Control System (MIMICS) software for the calculation of the osteointegration; they however failed to specify the methodology used to perform such calculations [27].

Despite the difficulties also highlighted by Maenhoudt et al. in defining the osteointegration rate, they reported a good integration rate that ranged from 25% to 82%. The highest integration rate was achieved in younger patients after 12 months from the implantation. It is now clear that future studies should put more effort into using an objective method of evaluation, such as the use of software, since the use of the only visual assessment, even if performed by experienced neuroradiologists, tends to under or over-estimate the effective rate of osteointegration [27].

A successful integration between the device and the cranial vault is not only dependent on the age of the patients, but other important factors affect the final results, such as the reason for which the decompressive craniectomy has been performed in the first place, but also the dimension of the skull defect and also the presence of other chronic condition, such as diabetes mellitus [31,32]. Besides those considerations, there are also intraoperative factors that have the potential to affect the final result. It has been shown that a tight and careful placement and fixation of the cranial prosthesis Is responsible for the reduction in the micromovements, allowing a more general stabilization; this could be further increased by the freshening of the edges by removing all the scar tissue that has been formed from the first surgery to maximize margin adherence [18]. Another technical nuance to take into consideration is the musculocutaneous flap that facilitates vascular ingrowth and allow also better preservation of the dura mater [23]. Post-gadolinium T1 MR images make it possible to visualize the presence of these small vessels inside the prosthesis [33].

The whole concept of osteointegration in cranial reconstruction is now changing: the philosophy behind the cranioplasty surgery is no longer the reparation of the opercula, but the generation of new tissue (Figure 1). The healing of the skull and the restoration of its integrity depends on both biological and mechanical requirements. The biological requirements depend on the presence of bone cells that are capable of supporting healing and an adequate vascular supply. The mechanical requirements obey the degree of rigidity and stability of the “craniolacunia–bone flap” system, which mainly depends on the surgical fixation technique [34].

## 4. Cement Hydroxyapatite

Hydroxyapatite cement is a set of pastes made of a calcium phosphate preparation. They are widely used in the surgical setting since they can be shaped intraoperatively by the attending surgeon [35]. The present scientific literature does not report noticeable toxic reactions, implants extruded, or wound infections. However, it has been suggested by some histologic examinations performed in those implants that there could be episodes of transient inflammatory response. In the study, it has been not reported any case of ì foreign body reaction [35].

Nowadays in neurosurgery, hydroxyapatite cement is especially used in for the reconstruction of the skull after skull base surgeries, in particular in the postoperative reconstruction after the translabyrinthine approaches (TLAs). TLAs are a series of skull base surgery approaches mainly used to resect skull base neoplasms. Recent evidence has suggested that the mix of such pastes with autologous fat grafts of the patients is responsible for the superior outcome in terms of postoperative complications. Concerning the clinical outcome, cement hydroxyapatite showed a postoperative complication rate similar to other reparative techniques [35]. Some studies supported the idea through the use of cement HA, there was a reduction in the length of stay (LOS) in the hospital and a reduction in episodes of CSF leaks. The satisfaction in using cement HA for the reconstruction after TLAs surgeries is not limited to the clinical outcome, but also to patients for the good cosmetic results [36]. such an option has resulted to be feasible also after other types of neurosurgical procedures, such as the suboccipital retrosigmoid [37].

Cement hydroxyapatite has been also used in pediatric patients. Despite the lack of large prospective studies, it seems as valid as in adult patients. It seems also that the use of computational simulations in the decision of the shape and size of the pore of the device implanted may alter the success of osteointegration [38].

## 5. Custom-Made Cranioplasty

In recent years, custom-made cranioplasties made of PHA have been widely used [7]. As per the name, these devices are prepared specifically for the patient. The cranioplasty is constructed based on a 3D CT-scan that the attending neurosurgeon gives to the manufacturer. With that, the company can construct a prototype that has to be analysed and accepted by the neurosurgeon before the production of the device that will be used. There has been also an increase in its use in recent years because of its well-known osteointegrative properties as well as not concerning risk of postoperative infection [25,26,28,30]. A recent multicentric European retrospective study, with the largest clinical series of patients treated with PHA cranioplasty published (494 patients) showed that this material has a 7.9% rate of complications with an explantation rate of 4.3%. The most common complications were infection (4.86%), hematomas (1.2%), fractures (1.01%), mobilization (0.4%) and scar retraction (0.4%) [6]. Fractures are not only one of the most common postoperative complications for this material but also quite concerning and its management remains a topic of discussion (Figure 2). The authors, through their multicentric experience, have been able to show that despite its relatively important prevalence, especially in the first phase of implantation, when there is still no osteointegration, only, a smaller portion of patients with fractures needed a surgical revision.

These data are more significant when the clinical outcome of porous hydroxyapatite is compared with other heterologous materials. Recently, a systematic review of the literature aiming to examine the association between the material of choice for cranial reconstruction and related complications to suggest the strategy treatment has been published. The authors of the study performed also a sub-analysis of postoperative complications only related to the material choice, such as postoperative infections, fractures, and prosthesis displacement. This analysis showed that patients implanted with porous hydroxyapatite presented a lower rate of postoperative infection and explantation after infections, compared to patients implanted with other heterologous materials included in the study (PMMA, PEEK and titanium). On the other side, hydroxyapatite patients reported a significantly higher rate of prosthesis displacement. Despite the great significance of these conclusions, further studies, in particular clinical trials, are needed since this systematic review lack a statistically correct analysis [7].

Generally speaking, from a clinical point of view, the PHA cranioplasty showed to be an effective option, at least as effective as the other heterologous materials present in the market, in terms of postoperative outcome and complications. The only exception is made for the prevalence of fracture. Despite an increased risk compared to other options available on the market because of its physical and biochemical properties, neurosurgeons should be able to differentiate between asymptomatic and symptomatic [39,40]. Only a small quantity of the fractures is effectively symptomatic and needs a surgical revision

## 6. Custom Made Cranioplasty in Children

Management of cranial reconstruction in the pediatric population remains an even more difficult task. The first difficulty derives from the imprecision in the definition of what should be considered pediatric but also the fact that the practical guidelines that are nowadays present do not consider the age of growth of the cranial vault [41]. Postoperative complications tend to be also more difficult to management and are present in higher rates across not only in autologous bone, where the rate of bone reabsorption has been reported to be as high as 50%, but also in heterologous materials [42].

The PHA has been used effectively in all different sets of age, up to children 1-year-old. A multicenter post-marketing surveillance study with 76 implants in 67 patients and a mean age 10.03 ± 1.72 years, with age stratification, almost equally distributed has been published. The mean follow-up was about 36 months. The postoperative complication rate was 14.5%. Failure rate, meaning the need for revision surgery, was 6.6%, which is statistically not superior to the explantation rate recorded in adults (two-sided 95%, CI 2.2–14.7%) [43].

A systematic review of the literature has been published in 2019 with the aim to analyse the current cranioplasty practice in children. The review consisted in 24 manuscripts, describing a total of 864 cranioplasty procedures, including both autologous and heterologous cranioplasties. The authors highlighted that the authors found that custom-made porous hydroxyapatite is the most commonly used material (26.3% of the total series), probably due to the fact that it is the only one being able to offer the osteointegrative properties, which are well adjusted in the pediatric population [44,45].

## 7. Conclusions

Porous hydroxyapatite is a valid bioceramic and its application should be further explored, not only for neurosurgical applications. For what concern its application for cranial reconstruction, this heterologous material seems to be at least as valid as other materials actually present on the market for what concerns the postoperative clinical outcome. Despite not being the ideal material for cranioplasty, compared to the other options, it presents the advantage of osteointegration.

## Figures and Tables

**Figure 1 ijms-23-05434-f001:**
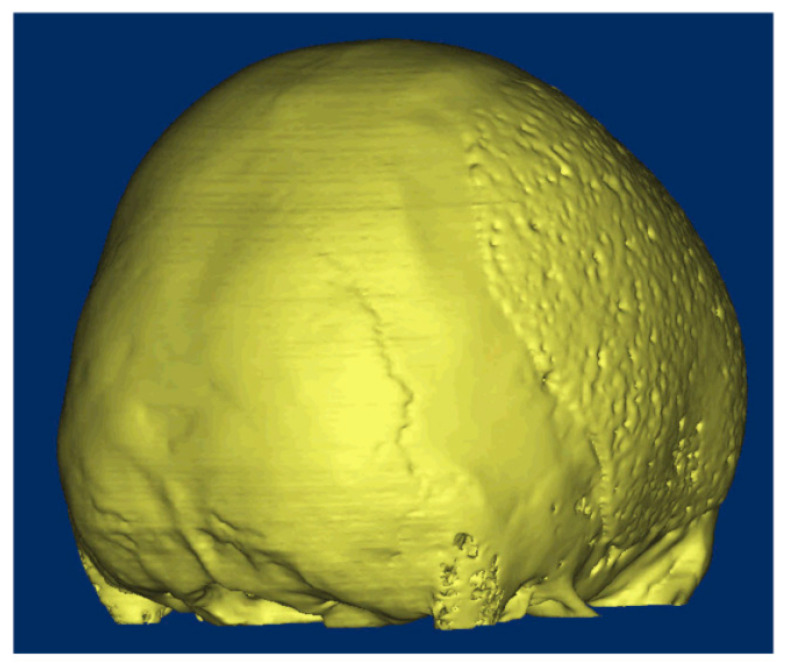
A 16-year-old patient came to our attention after a traumatic brain injury (TBI) for which a decompressive craniectomy (DC) was required. After 4 months, the attending neurosurgeons decided that the patient was stable, and it could be implanted. The surgical procedure and the postoperative follow-up were uneventful. At 8 months follow-up, the CT-scan showed a complete osteointegration.

**Figure 2 ijms-23-05434-f002:**
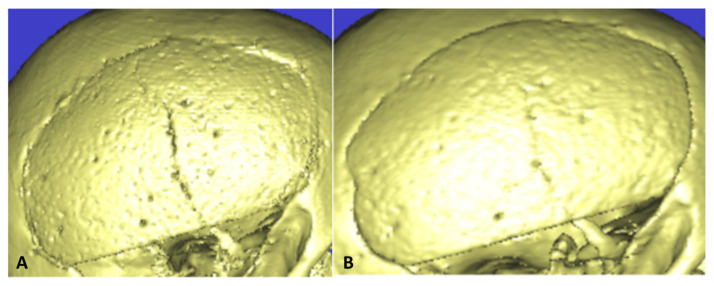
A female patient aged 32 years was admitted in the neurosurgical service after a decompressive craniectomy for a stroke of the middle cerebral artery (MCA). The patient was implanted with the autologous bone 6 months after the first surgery, but she underwent a revision surgery for bone reabsorption. After a multidisciplinary discussion, the neurosurgeon decided to implant the patient with a PHA cranioplasty. The surgical implantation was uneventful. At the first follow-up of three months (**A**), there was a sign on the CT-scan of an asymptomatic fracture. The neurosurgeon decided to not operate. A follow-up after 7 months showed a complete bone healing without surgical intervention (**B**).

## Data Availability

All data are available on the medical databases.

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
