# Peer review of "From Reparative Surgery to Regenerative Surgery: State of the Art of Porous Hydroxyapatite in Cranioplasty"

_ijms, 2022, doi:10.3390/ijms23105434_

Round 1

Reviewer 1 Report

This paper reviews the use of porous hydroxyapatite (PHA) in cranial reconstruction for a regenerative method rather than a reparative one. The authors discuss the characteristics of PHA and present previous published studies. The authors deal with an interesting topic, but the manuscript is difficult to follow. Some suggestions and concerns are listed below that could improve the manuscript. 

  • In the section of “2. Molecular Composition”, there are a couple of paragraphs that discuss physical factors such as porosity and pore interconnectiviey.  These factors are not related with the molecular composition of PHA. 
  • In section of “3. Radiological Evaluation and Osteointegration”, the authors discuss the osteointegration results from PHA implant.  The comparison of osteointegration among autologous bone, other heterologous materials (PMMA, PEEK) and PHA would be interesting.
  • On page 3, the authors mention that “The whole concept of osteointegration in cranial reconstruction is not changing the philosophy behind the cranioplasty surgery is not anymore the reparation of the opercula, but the generation of a new tissue (Figure 1).” To support this declaration, the authors need to discuss biological response from the PHA in comparison with other prostheses including autologous bone, PMMA, PEEK and titanium.
  • Sections of “5. Custom-Made Cranioplasty” and “6. Custom Made Cranioplasty in Children”: Some explanations about the preparation method and characteristics of “custom-made” PHA are needed, as the authors discuss the other type of HAP in the previous section “4. Cement Hydroxyapatite”.
  • There are two figures in Figure 2.  The figure caption needs a detailed description about them.
  • References need to be properly referenced, especially in the section of “3. Radiological Evaluation and Osteointegration”. 
  • The references should be numbered in order of appearance. Ref 37 and 38 appeared after ref 15.
  • The manuscript should be proofread carefully. 

Author Response

Dear Reviewer,

we would like to thank you for your time and effort in revising our submission. You can find below our response to your comment in a point-like manner. We hope that you find our revised version worth of publication.

“In the section of “2. Molecular Composition”, there are a couple of paragraphs that discuss physical factors such as porosity and pore interconnectiviey.  These factors are not related with the molecular composition of PHA”
The paragraph has been completely re-written in order to make it more appropriate to the focus of the paper

“In section of “3. Radiological Evaluation and Osteointegration”, the authors discuss the osteointegration results from PHA implant.  The comparison of osteointegration among autologous bone, other heterologous materials (PMMA, PEEK) and PHA would be interesting.”
Despite agreeing with the reviewer that a comparison will make the discussion more interesting, it could be not done, since the osteointegration has been seen only in PHA and not in other heterologous materials. This has been specified in the paper

On page 3, the authors mention that “The whole concept of osteointegration in cranial reconstruction is not changing the philosophy behind the cranioplasty surgery is not anymore the reparation of the opercula, but the generation of a new tissue (Figure 1).” To support this declaration, the authors need to discuss biological response from the PHA in comparison with other prostheses including autologous bone, PMMA, PEEK and titanium.
We agree with the reviewer. We have specify in the paper that there are strong scientific evidences that the porous HA implant significantly promotes early bone ingrowth at the pre-generated defective region, and early fixation at the bone-implant interface

“Sections of “5. Custom-Made Cranioplasty” and “6. Custom Made Cranioplasty in Children”: Some explanations about the preparation method and characteristics of “custom-made” PHA are needed, as the authors discuss the other type of HAP in the previous section “4. Cement Hydroxyapatite”.”
We agree with the reviewer. We explained the preparation method and the characteristics of custom-made PHA in the Section 5

There are two figures in Figure 2.  The figure caption needs a detailed description about them.”

We provided a more detailed captions for both the figures

“References need to be properly referenced, especially in the section of “3. Radiological Evaluation and Osteointegration”. 

The section has been improved and we added more references to support our statements

“The references should be numbered in order of appearance. Ref 37 and 38 appeared after ref 15.”
We have tried to number in order of appearance all the new references included in the paper

“The manuscript should be proofread carefully. “
we have asked a native English speaker medical doctor to revise and proofread our paper

Reviewer 2 Report

The article ‘From Reparative Surgery to Regenerative Surgery: State of the Art of Porous Hydroxyapatite in Cranioplasty’ is a mini review summarizing applications of hydroxyapatite-based bioceramics for cranial reconstruction, demonstrating its clinical properties and limitations. The review is interesting and may be useful for the biomaterials community however, I have a few comments:

  1. The introduction is too short. The authors briefly wrote about metals polymers used in cranioplasty. I suggest the authors at least list the advantages and disadvantages of other biomaterials used in CP. More literature in references, regarding those materials would be useful.
  2. ‘This has been already widely proved in different types of applications, such as in orthopaedic, dental, and maxillofacial surgery [9].’ This sentence is true, but the number of references is not sufficient. The material is widely used, so the number of references should be larger.
  3. ‘Hydroxyapatite can be employed in forms such as powders, injectable pastes, porous blocks, or beads to fill bone defects or voids which can arise when large sections of bone have had to be removed (e.g. bone cancers) or when bone augmentations are required (e.g. maxillofacial reconstructions or dental applications). [10]’ The number of references is not sufficient to support this sentence.
  4. ‘Several studies investigated the minimum pore size required to regenerate mineralized bone [12,13].’ The authors did not write any details- so what should be the optimal size of pores?
  5. In my opinion it should be mentioned that porosity also affects the mechanical strength of the implant material.

Author Response

Dear Reviewer,

we would like to thank you for your time and effort in revising our submission. You can find below our response to your comment in a point-like manner. We hope that you find our revised version worth of publication.

“The introduction is too short. The authors briefly wrote about metals polymers used in cranioplasty. I suggest the authors at least list the advantages and disadvantages of other biomaterials used in CP. More literature in references, regarding those materials would be useful.”

We have re-written our introduction, briefly discussing the other main other heterologous material and the fundamental differences with porous hydroxyapatite

“‘This has been already widely proved in different types of applications, such as in orthopaedic, dental, and maxillofacial surgery [9].’ This sentence is true, but the number of references is not sufficient. The material is widely used, so the number of references should be larger.”

We have added several citation to back our statement.

“‘Hydroxyapatite can be employed in forms such as powders, injectable pastes, porous blocks, or beads to fill bone defects or voids which can arise when large sections of bone have had to be removed (e.g. bone cancers) or when bone augmentations are required (e.g. maxillofacial reconstructions or dental applications). [10]’ The number of references is not sufficient to support this sentence.”

Also in this case, we have added other references to back the sentence

“‘Several studies investigated the minimum pore size required to regenerate mineralized bone [12,13].’ The authors did not write any details- so what should be the optimal size of pores?”

As requested, we added more details on the topic in the manuscript, in particular that the size of osteoblasts is on the order of 10–50 μm, however osteoblasts prefer larger pores (100–200 μm) for regenerating mineralised bone after implantation. This allows macrophages to infiltrate, eliminate bacteria and induce the infiltration of other cells involved in colonisation, migration and vascularisation in vivo. Whereas a smaller pore size (<100 μm) is associated with the formation of non-mineralised osteoid or fibrous tissue

“In my opinion it should be mentioned that porosity also affects the mechanical strength of the implant material.”

We would like to thank the reviewer for raising such concerning point. We have added in the manuscript what are the consequences, also from the clinical point of view of a larger number of pores and how their affect the general structure of the device.

Round 2

Reviewer 1 Report

Thank the authors for revising the manuscript in response to the reviewers' comments.  The manuscript has been significantly improved.  The suggestions and concerns are listed below.

  • The authors should consider changing of the section title “2. Molecular Composition”. In addition to the molecular characteristics of PHA, this section covers the physicochemical properties of PHA implants such as porosity, interconnectivity and mechanical strength.  
  • On page 2, ref #41 appears after #10. References should be numbered consecutively in the order they appear throughout the paper.
  • In a paragraph of “From the clinic point view, it has been … by recent multicentric study [38].” on page 3, authors added references #37 and #38 to support PHA’s ability to prevent biofilm formation. In ref #38, PHA was used with antibiotics, so the results in ref #38 do not support the authors’ claim. It needs some explanation and/or other references.
  • Figure 2 needs a caption that explains what the left and right images are. I assume that the left image is from 3-month follow-up and the right image from 7-month follow-up.
  • There are many grammar errors throughout the manuscript.

Author Response

Dear Reviewer,

we would like to thank you for your time and effort in revising our submission. You can find below our response to your comment in a point-like manner. We hope that you find our revised version worth of publication.

“Thank the authors for revising the manuscript in response to the reviewers' comments.  The manuscript has been significantly improved.  The suggestions and concerns are listed below. The authors should consider changing of the section title “2. Molecular Composition”. In addition to the molecular characteristics of PHA, this section covers the physicochemical properties of PHA implants such as porosity, interconnectivity and mechanical strength.”
As suggested by the Reviewer, we have changed the title of the section

On page 2, ref #41 appears after #10. References should be numbered consecutively in the order they appear throughout the paper.”
We agree with the Reviewer, we have revised the order of the references

“In a paragraph of “From the clinic point view, it has been … by recent multicentric study [38].” on page 3, authors added references #37 and #38 to support PHA’s ability to prevent biofilm formation. In ref #38, PHA was used with antibiotics, so the results in ref #38 do not support the authors’ claim. It needs some explanation and/or other references.”
We thank the reviewer for the observation. We have reformulated the sentence and implemented the references

Figure 2 needs a caption that explains what the left and right images are. I assume that the left image is from 3-month follow-up and the right image from 7-month follow-up.”
We have re-formatted the figure 2 and we improved the caption, explain the images

There are many grammar errors throughout the manuscript.”
We have asked a native medical doctor to revise the quality of our manuscript

Round 3

Reviewer 1 Report

The reviewer's comments were adequately addressed by the authors.